# Predictive value of ultrasound doppler parameters in neoadjuvant chemotherapy response of breast cancer: Prospective comparison with magnetic resonance and mammography

**Livia Conz[1,2], Rodrigo Menezes Jales[3], Maira Teixeira Dória[4], Isabelle Melloni[3], Carla Andries Cres Lyrio[3], Carlos Menossi[3], Sophie Derchain[1,2], Luís Otávio Sarian[1,2]***

1 Department of Obstetrics and Gynecology, State University of Campinas (Unicamp), Campinas, São Paulo, Brazil, 2 Division of Gynecologic and Breast Oncology, Women's Hospital (CAISM), Unicamp, Campinas, São Paulo, Brazil, 3 Imaging Sector, Women's Hospital (CAISM), Unicamp, Campinas, São Paulo, Brazil, 4 Department Obstetrics and Gynecology, Federal University of Parana (UFPR), Curitiba, Parana, Brazil

* sarian@unicamp.br

## Abstract

### Background

Neoadjuvant chemotherapy (NACT) is a treatment option for breast cancer patients that allows for the assessment of tumor response during treatment. This information can be used to adjust treatment and improve outcomes. However, the optimal imaging modalities and parameters for assessing tumor response to NACT are not well established.

### Methods

This study included 173 breast cancer patients who underwent NACT. Patients were imaged with ultrasound (US), mammography (MMG), and magnetic resonance imaging (MRI) at baseline, after two cycles of NACT, and before breast surgery. US parameters included lesion morphology, Doppler variables, and elastography measurements. MMG and MRI were evaluated for the presence of nodules and tumor dimensions. The pathological response to NACT was determined using the residual cancer burden (RCB) classification.

### Results

The US parameter with the highest power for predicting pathological complete response (pCR) was shear wave elastography (SWE) maximum speed inside the tumor at baseline. For nonluminal tumors, the end diastolic velocity measured by US after two cycles of NACT showed the highest predictive value for pCR. Similarly, SWE maximum speed after two cycles of NACT had the highest discriminating power for predicting RCB-III in luminal tumors, while the same parameter measured at baseline was most predictive for nonluminal tumors.

**Data Availability Statement:** All relevant data are within the paper and its Supporting Information files.

**Funding:** The author(s) received no specific funding for this work.

**Competing interests:** The authors have declared that no competing interests exist.

## Conclusions

This study provides evidence that mid-treatment Doppler US and other imaging modalities can be used to predict the response to NACT in breast cancer patients. Functional parameters, such as blood flow velocities and SWE measurements, demonstrated superior predictive value for pCR, while morphological parameters had limited value. These findings have implications for personalized treatment strategies and may contribute to improved outcomes in the management of breast cancer.

## Introduction

Neoadjuvant chemotherapy (NACT) for the treatment of breast cancer has similar results to adjuvant chemotherapy in terms of overall survival and disease-free survival [1,2]. Furthermore, it allows the in vivo monitoring of the tumor response to chemotherapy [3], in addition to favoring the indication of conservative surgery [4]. Finally, complete anatomopathological response to NACT is an important marker of long-term disease-free survival and overall survival [2]. The anatomopathological description of the tumor response to NACT through the residual cancer burden (RCB) quantifies the residual disease at the surgical site [5].

However, the assessment of tumor response to neoadjuvant treatment by clinical evaluation or imaging tests is still hampered by different factors, such as desmoplastic reaction, necrosis, fibrosis and tumor fragmentation, in addition to the possibility of the persistence of the component in situ [6].

Mammography (MMG) is the most performed test in the staging of breast cancer and has good accuracy in determining the size of the tumor [7]. However, its effectiveness in assessing the response to NACT is relatively low and varies from 26% to 31.7% [6] In a retrospective study with 446 patients undergoing NACT, MMG showed a sensitivity of 94% and a specificity of 50% when compared with the histopathological evaluation of the tumor [7]. Magnetic resonance imaging (MRI) has high specificity, between 89% and 92%, in predicting complete anatomopathological response (pCR) [8]. On the other hand, the sensitivity is relatively lower, between 56% and 70% [9]. Thus, although up to 60% of patients show pCR on the surgical specimen, only 20% of patients usually show a complete response to MRI [8,9].

Ultrasonography (US) has some benefits in breast workup compared to MRI, such as reduced cost, independence from contrast media, and safety and availability for serial examinations [10]. In addition, the accuracy of bright-mode US (B-mode), which assesses lesion morphology, may be comparable to that obtained by MRI in assessing the response to NACT [11].

Early diagnosis of nonresponse to chemotherapy would make it possible to suspend or modify chemotherapy, save limited financial resources, deprive patients of unnecessary side effects and expedite surgical treatment [12]. On the other hand, in cases in which the association of clinical examination and different imaging resources present high predictive values for a complete response to chemotherapy, patients may be encouraged by the assistant team to strictly comply with the chemotherapy treatment [13]. Furthermore, in these cases, patients in whom surgical treatment may be considered unnecessary may be selected for controlled research projects.

## Methods

### Patient selection

Between January 2017 and December 2019, a total of 173 (178 breast tumors in total due to bilateral lesions) breast cancer patients referred to NACT were included. Recruitment began

on January 31, 2017, and the last accrual occurred on November 30, 2019. Follow-up data in the present analysis covers the period through August 8, 2023. All patients were informed of the study objectives and signed a written informed consent form. This study did not accept minors. The Ethics Committee of Campinas State University- UNICAMP (CEP Protocol #59361616000005404) has approved the study on October 31, 2016.

After agreeing to enroll, patients were interviewed for key clinical and epidemiological characteristics by one of the main investigators. Next, on the same day, patients underwent MMG and a breast US examination. After undergoing two NACT cycles, patients underwent a new round of US, MMG and MRI. At that point in time, clinical events related to NACT, and disease progression were logged by the researchers. Next, patients completed their NACT cycles and underwent the last round of US, MMG and MRI. Following the standard therapeutic protocols, patients were further treated with breast surgery (either mastectomy, n = 68, or conservative surgery, n = 105). The pathological assessment of the surgical specimens was used to define the pathological response to NACT (see details below).

## Ethics

All patients were informed of the study objectives and signed a written informed consent form. This study did not accept minors. The Ethics Committee of Campinas State University (CEP Protocol #59361616000005404) has approved the study on October 31, 2016.

## Ultrasound

All ultrasound examinations were performed at the Medical Imaging Sector of our Institution using Acuson S2000 equipment (Siemens®, Munich, Germany) equipped with a high-resolution, multifrequency linear transducer capable of B-mode, Doppler and shear wave elastography (SWE) technologies. US was performed three times: at baseline, after two NACT cycles and before breast surgery.

All examinations were performed by a trained researcher (CM) under the supervision of an experienced ultrassonographyst (RMJ—>20 years of uninterrupted practice in breast US). For B-mode ultrasonography, performed with a 14 MHz linear transducer, the presence or absence of an ultrasonographic nodule was evaluated. When present, the shape, margins, echotexture and orientation of the nodule were described. Linear measurements of the nodule (longitudinal, anteroposterior and transverse) were performed. After that, the Doppler variables were evaluated, also with the 14 MHz linear transducer. Initially, the amplitude Doppler was assessed, which will identify the presence or absence of flow within or adjacent to the nodule. In cases with the presence of flow inside or adjacent to the nodule, it was subjectively quantified by the researcher as weak, moderate or intense. The quantitative variables were then measured using spectral Doppler. We assessed peak systolic velocity (PS), end diastolic velocity (ED), mean flow velocity (MF), pulsatility index (PI), and resistive index (RI) in the longitudinal plane of the arteries. Additionally, in the longitudinal plane, we evaluated the maximum velocity (Vmax) of the veins. The Doppler indicators were automatically calculated using the absolute Doppler velocimetry obtained by insonating the central portion of the breast lesions, as perceived by the sonographer. The Doppler variables thus refer to the largest vessels identified in each tumor, as first identified using color Doppler and then measured using spectral Doppler. The only correction applied was for the angle of insonation, for values <60 degrees.

Finally, elastography was performed only in cases where there was a B-mode US nodule. In this plane, the maximum US propagation velocity of the nodule was measured and quantified in meters per second. We took two measurements for each lump, one in the central region of the tumor, and another on parenchyma surrounding the tumor.

## Mammography

MMG examinations were performed at the radiology section of our Institution. Mammomat equipment (Siemens®, Munich, Germany) was used to produce the mammographic studies. For each study, two views were obtained: mid-lateral-oblique and craniocaudal. The images were processed by the technologist CR (Computing Radiology) and archived in a PACS (Picture Archiving in Communication System) system available at our Institution. The presence or absence of a mammographic nodule was evaluated. When present, the shape (largest diameter, largest height), margins, echotexture and orientation of the nodule were evaluated. Linear measurements of the nodule (longitudinal, anteroposterior and transverse) were performed three times: at baseline, after two NACT cycles and before breast surgery.

## Magnetic resonance imaging

MRI was performed at two different times: baseline and preoperative. All MRI exams were performed with the same pre- and postcontrast image acquisition protocol, according to the consensus of the radiology services of our Institution. Exams were performed using General Electric Signa HDxt® 1.5 Tesla equipment (GE, Milwaukee, WI, United States of America). Exams were performed using Achieva 1.5 Tesla equipment (Philips, Eindhoven, Netherlands).

## Pathological assessment

The surgical specimen was analyzed according to the routines of the pathology services at our Institution. The "Residual Cancer Burden" (RCB) classification of residual cancer [14] was included in the pathology report. An RCB score is determined using information on the size of the tumor and the extent of tumor cells in the breast and axillary lymph nodes after neoadjuvant therapy. The higher the RCB score, the more residual [14] invasive breast cancer there is in the breast and lymph nodes:

RCB-0 = No residual invasive breast cancer (same as pathologic complete response- pCR)

RCB-I = Small amount of residual invasive breast cancer

RCB-II = Moderate amount of residual invasive breast cancer

RCB-III = Extensive (a lot of) residual invasive breast cancer

Complete anatomopathological response (pCR) in the surgical specimen was the gold standard: absence of residual disease at the tumor site evaluated by the anatomopathological study of the surgical specimen according to standard practices, later categorized by the researcher as yes or no. Extensive residual disease in the surgical specimen: presence of extensive residual disease in the tumor site evaluated by the anatomopathological study of the surgical specimen.

## Statistical analysis

All statistical calculations were performed using the R environment for statistical computing. Confidence levels were set at 5%. First, we compared the key clinical and epidemiological features of the patients as related to their response to NACT, considering the extremes of the clinical spectrum. Next, using receiver operating characteristic (ROC) curve analysis, we calculated the optimal cutoff point for the US, MMG, and MRI parameters, as measured before or after two NACT cycles, at diagnosing either pCR or RCB-III after completion of NACT. Then, using the optimal cutoff points for each parameter at either the baseline or after two NACT cycles, we calculated the likelihood ratio and then the posttest probabilities of a patient achieving either pCR or RCB-III after completing their neoadjuvant treatment. Fagan's

nomogram depictions were produced for the 4 test parameters with the highest differences between the posttest probabilities (absolute value of positive posttest probability minus modulus of the negative posttest probability).

## Results

**Table 1** shows the key clinical and pathological features of the patients as related to the extremes of the clinical response spectrum (either RCB-III or pCR) after NACT. Of the 173 patients, 41 (23%) evolved with RCB-III and 30 (17%) with pCR after NACT. A significantly higher proportion of patients with invasive lobular carcinoma (ILC; 66.7%) had RCB-III as the outcome after NACT compared to patients with invasive ductal carcinomas (IDC, 21.1%), p = 0.02. In addition, patients with luminal A tumors also evolved with RCB-III (45.2%) more frequently than their counterparts with luminal B (22.1%), HER 2 (5.9%) or triple-negative (16.7%) tumors; p = 0.007. Similarly, patients with stage III tumors had a significantly higher proportion of RCB-III responses (37.3%) than their counterparts with either stage II (17.8%) or stage I (zero percent) tumors. None of the studied patients' or pathological features was associated with pCR.

**Table 2A and 2B** list the different imaging modalities and imaging parameters according to the difference between the posterior probabilities (DPP) of pCR in luminal (**Table 2A**) and nonluminal (**Table 2B**) tumors after NACT. Parameter thresholds were calculated using ROC analysis (please refer to Statistical Analysis). The larger the DPP for patients with a positive (above threshold) test versus that for patients with a negative (below threshold) test, the greater the discriminating power of that test parameter. The largest DPP was obtained by measuring SWE maximum speed inside the tumor at baseline (i.e., before the NACT start); at a threshold of 9.99 cm (about 3.93 in/s), a positive test yielded a posterior probability of pCR of 16.33%, versus 3.88% for a negative test. The second largest DPP (9.65%) was obtained by measuring the resistive index using Doppler ultrasound at baseline, followed by the systolic/diastolic ratio at baseline (DPP = 9.41%). It is worth noting that negative DPP values merely reflect the fact that patients with a positive (i.e., above threshold) test have a lower posterior probability than those with a negative test; i.e., that parameter is inversely associated with the probability of a given patient evolving with pCR. **Table 2B** shows the likelihood analysis for the prediction of pCR in patients with nonluminal tumors (either triple-negative or nonluminal HER2 positive). **Table 2A** lists the test parameters, performed either at baseline or after 2 NACT cycles, from the highest to the lowest absolute difference between posttest probabilities. The highest DPP was obtained by measuring the end diastolic velocity (cutoff = 1.9 cm -about 0.75 in/s); positive posttest probability = 47.4%, negative = 11.5%; difference = 35.9%, measured using ultrasound, after 2 NACT cycles. A slightly inferior difference (29.3%) was obtained with the same parameter but when performed at baseline.

**Table 3A and 3B** depict the results of analyses similar to those described for Table 2A and 2B but with RCB-III as the outcome of interest. For luminal tumors (Table 3A), the best prediction was obtained using shear wave elastography maximum speed inside the tumor, as measured after 2 NACT cycles (cutoff = 9.06 cm -about 3.57 in/s); positive posttest probability = 33.06%, negative = 17.81%; difference = 15.25%.

Finally, in **Table 3B,** we present the same analyses for nonluminal tumors, which yielded much larger DPP values compared to the analyses performed for patients with luminal tumors. The largest DPP was obtained by measuring the baseline values of the largest height and largest tumor diameters using MRI (cut off = 42 and 51 mm -about 2.01 in/s, respectively, difference = 45.74% and 28.57%, respectively). It is worth noting that for patients with the largest

**Table 1. Key clinical and pathological features related to the response to neoadjuvant chemotherapy (NACT): Residual cancer burden–III (RCB-III) and pathological complete response (pCR).**

| Characteristic | n[%] | RCB-III | p value | pCR | p value |
|---|---|---|---|---|---|
| *Age* | | n[%] | | n[%] | |
| ≥50 years | 86 | 15 [17.4%] | 0.08 | 15 [17.4%] | 1 |
| <50 years | 87 | 26 [29.9%] | | 15 [17.2%] | |
| *BMI [kg/m2]* | | | | | |
| <25 | 53 | 11 [20.8%] | 0.56 | 10 [18.9%] | 0.89 |
| 25 to 30 | 46 | 9 [19.6%] | | 7 [15.2%] | |
| ≥30 | 58 | 16 [27.6%] | | 10 [17.2%] | |
| *Pathology* | | | | | |
| IDC | 161 | 34 [21.1%] | 0.02 | 29 [18%] | 0.51 |
| ILC | 6 | 4 [66.7%] | | 0 [0%] | |
| Other | 5 | 2 [40.0%] | | 1 [20%] | |
| *Molecular subtype* | | | | | |
| Luminal A (RE/RP positive, KI67<20%) | 31 | 14 [45.2%] | 0.007 | 0 [0%] | 0.98 |
| Luminal B (RE/RP positive, KI67>20%) | 95 | 21 [22.1%] | | 13 [13.7%] | |
| HER 2 (HER 2 positive) | 17 | 1 [5.9%] | | 9 [52.9%] | |
| Triple-Negative (RE/RP/HER 2 negative) | 30 | 5 [16.7%] | | 8 [26.7%] | |
| *Clinical stage* | | | | | |
| I | 7 | 0 [0%] | 0.005 | 1 [14.3%] | 0.83 |
| II | 107 | 19 [17.8%] | | 20 [18.7%] | |
| III | 59 | 22 [37.3%] | | 9 [15.3%] | |
| *Axilla status* | | | | | |
| N0 | 69 | 11 [15.9%] | 0.07 | 7 [10.1%] | 0.06 |
| N1 or above | 104 | 30 [28.8%] | | 23 [22.1%] | |
| *Menopause* | | | | | |
| No | 89 | 17 [19.1%] | 0.19 | 13 [14.6%] | 0.43 |
| Yes | 84 | 24 [28.6%] | | 7 [20.2%] | |
| *Hormonal therapy* | | | | | |
| Never | 93 | 24 [25.8%] | 0.31 | 15 [16.1%] | 1 |
| Current or past | 17 | 7 [41.2%] | | 3 [17.6%] | |
| *Smoking* | | | | | |
| Never | 124 | 26 [21%] | 0.18 | 22 [17.7%] | 0.25 |
| Past smoker | 30 | 11 [36.7%] | | 7 [23.3%] | |
| Current smoker | 19 | 4 [21.1%] | | 1 [5.3%] | |
| *First degree relative with breast or ovarian cancer* | | | | | |
| No | 145 | 35 [24.1%] | 1 | 25 [17.2%] | 1 |
| Yes | 26 | 6 [23.1%] | | 5 [19.2%] | |

BMI = body mass index; kg/m2 = kilogram/square meter; IDC = invasive ductal carcinoma; ILC = invasive lobular carcinoma; RE = estrogen receptor;

RP = progesterone receptor; HER2 = human epidermal growth factor receptor-type 2.

tumor height >42 mm (about 1.65 in), the probability of the patient having RCB-III after NACT sits at 50%, contrasted to only 3.95% for those patients with smaller tumors at baseline.

*Fig 1* presents the flowchart (STARD) illustrating the study design and patient progression throughout the study. *Fig 2* displays Fagan's representation of prior and posterior probabilities in the best-case scenarios for predicting either pCR or RCB-III in luminal and nonluminal tumors. The red lines indicate the prior and posterior probabilities when the test result is below the threshold, while the green lines connect the probabilities when the test result is

**Table 2. A. Performance of individualized imaging parameters, performed either before or during neoadjuvant chemotherapy (NACT), for the diagnosis of pathological complete response (pCR) in luminal tumors, ranked by the difference between posterior probabilities at optimal exam cutoffs. B. Performance of individualized imaging parameters, performed either before or during neoadjuvant chemotherapy [NACT], for the diagnosis of pCR in nonluminal tumors, ranked by the difference between posterior probabilities at optimal exam cutoffs.**

| Imaging Modality | Test | Moment performed | Cutoff | Posterior probability of pCR if above cutoff | Posterior probability of pCR if below cutoff | Difference between posterior probabilities |
|---|---|---|---|---|---|---|
| SWE | Shear wave maximum speed inside [cm/s] | a) Baseline | 9.99 | 16.33% | 3.88% | 12.44% |
| Ultrasound | Resistive Index | a) Baseline | 0.85 | 16.30% | 6.65% | 9.65% |
| Ultrasound | Systolic/diastolic ratio index | a) Baseline | 6.71 | 16.45% | 7.04% | 9.41% |
| SWE | Shear wave maximum speed out [cm/s] | b) After 2AC cycles | 3.41 | 14.57% | 5.50% | 9.07% |
| Ultrasound | Pulsatility Index | a) Baseline | 1.85 | 16.50% | 7.51% | 8.99% |
| Ultrasound | End diastolic velocity [cm/s] | b) After 2AC cycles | 2.7 | 14.28% | 8.08% | 6.21% |
| MRI | Largest diameter [mm] | b) After 2AC cycles | 17 | 9.22% | 15.10% | -5.87% |
| Ultrasound | Pulsatility Index | b) After 2AC cycles | 1.57 | 13.87% | 8.21% | 5.66% |
| Ultrasound | Peak Systolic [cm/s] | b) After 2AC cycles | 8.3 | 12.44% | 6.82% | 5.62% |
| MRI | Largest diameter [mm] | a) Baseline | 39 | 14.05% | 8.67% | 5.37% |
| Ultrasound | Peak Systolic [cm/s] | a) Baseline | 11.2 | 13.15% | 7.80% | 5.36% |
| Ultrasound | Largest diameter [mm] | b) After 2AC cycles | 16.4 | 12.05% | 6.98% | 5.07% |
| Mammography | Largest height [mm] | b) After 2AC cycles | 15 | 12.47% | 7.43% | 5.04% |
| Ultrasound | Systolic/diastolic ratio index | b) After 2AC cycles | 4.13 | 12.52% | 7.63% | 4.89% |
| MRI | Largest height [mm] | b) After 2AC cycles | 11 | 9.37% | 14.10% | -4.72% |
| Ultrasound | Largest height [mm] | b) After 2AC cycles | 10 | 11.76% | 7.32% | 4.45% |
| Ultrasound | Largest height [mm] | a) Baseline | 15.2 | 12.00% | 7.84% | 4.16% |
| Mammography | Largest diameter [mm] | a) Baseline | 31 | 12.31% | 8.68% | 3.64% |
| Ultrasound | Resistive Index | b) After 2AC cycles | 0.76 | 11.83% | 8.21% | 3.62% |
| Mammography | Largest height [mm] | a) Baseline | 19 | 11.11% | 8.89% | 2.22% |
| Mammography | Largest diameter [mm] | b) After 2AC cycles | 19 | 10.99% | 9.07% | 1.92% |
| Ultrasound | Largest diameter [mm] | a) Baseline | 26 | 10.81% | 9.62% | 1.20% |
| Ultrasound | End diastolic velocity [cm/s] | a) Baseline | 2.1 | 10.94% | 9.76% | 1.18% |
| MRI | Largest height [mm] | a) Baseline | 18 | 10.01% | 10.98% | -0.97% |

*(Continued)*

**Table 2.** (Continued)

| Imaging Modality | Test | Moment performed | Cutoff | Posterior probability of pCR if above cutoff | Posterior probability of pCR if below cutoff | Difference between posterior probabilities |
|---|---|---|---|---|---|---|
| SWE | Shear wave maximum speed inside [cm/s] | b) After 2AC cycles | 8.75 | 10.06% | 10.89% | -0.83% |
| Imaging Modality | Test | Moment performed | Cutoff | Posterior probability of pCR if above cutoff | Posterior probability of pCR if below cutoff | Difference between posterior probabilities |
| Ultrasound | End diastolic velocity [cm/s] | b) After 2AC cycles | 1.9 | 47.43% | 11.53% | 35.90% |
| Ultrasound | End diastolic velocity [cm/s] | a) Baseline | 1.2 | 42.75% | 13.43% | 29.32% |
| Ultrasound | Peak Systolic [cm/s] | b) After 2AC cycles | 9.3 | 49.43% | 25.79% | 23.64% |
| SWE | Shear wave maximum speed out [cm/s] | b) After 2AC cycles | 3.35 | 23.61% | 45.18% | -21.57% |
| MRI | Largest diameter [mm] | b) After 2AC cycles | 33 | 49.07% | 28.64% | 20.42% |
| Mammography | Largest diameter [mm] | a) Baseline | 38 | 26.15% | 45.95% | -19.79% |
| Ultrasound | Peak Systolic [cm/s] | a) Baseline | 11.8 | 45.79% | 27.95% | 17.84% |
| Ultrasound | Shear wave maximum speed inside [cm/s] | a) Baseline | 9.6 | 30.49% | 46.22% | -15.73% |
| Ultrasound | Systolic/diastolic ratio index | a) baseline | 5.6 | 40.06% | 30.29% | 9.77% |
| MRI | Largest diameter [mm] | a) Baseline | 40 | 30.76% | 40.43% | -9.67% |
| MRI | Largest height [mm] | a) Baseline | 29 | 30.76% | 40.43% | -9.67% |
| Mammography | Largest height [mm] | b) After 2AC cycles | 24 | 30.36% | 39.53% | -9.18% |
| Ultrasound | Systolic/diastolic ratio index | b) After 2AC cycles | 3.63 | 33.27% | 41.60% | -8.33% |
| Ultrasound | Largest diameter [mm] | b) After 2AC cycles | 23 | 32.58% | 40.64% | -8.06% |
| Ultrasound | Largest diameter [mm] | a) baseline | 27 | 33.60% | 41.46% | -7.87% |
| SWE | Shear wave maximum speed inside [cm/s] | b) After 2AC cycles | 9.4 | 40.48% | 32.69% | 7.78% |
| Ultrasound | Resistive Index | a) Baseline | 0.82 | 38.30% | 32.56% | 5.74% |
| Ultrasound | Pulsatility Index | a) Baseline | 1.6 | 38.78% | 33.05% | 5.73% |
| Mammography | Largest height [mm] | a) Baseline | 19 | 35.05% | 39.31% | -4.25% |
| Ultrasound | Largest height [mm] | a) Baseline | 19 | 34.69% | 38.93% | -4.24% |
| Ultrasound | Pulsatility Index | b) After 2AC cycles | 1.28 | 37.56% | 34.27% | 3.29% |
| MRI | Largest height [mm] | b) After 2AC cycles | 20 | 34.86% | 37.58% | -2.72% |
| Ultrasound | Largest height [mm] | b) After 2AC cycles | 20.1 | 34.28% | 36.98% | -2.70% |
| Ultrasound | Resistive Index | b) After 2AC cycles | 0.68 | 36.68% | 34.27% | 2.41% |
| Mammography | Largest diameter [mm] | b) After 2AC cycles | 31 | 35.27% | 36.76% | -1.49% |

MRI = magnetic resonance imaging; SWE = shear wave elastography; AC = adriamycin [doxorubicin] plus cyclophosphamide.

**Table 3. A. Performance of individualized imaging parameters, performed either before or during neoadjuvant chemotherapy [NACT], for the diagnosis of residual cancer burden-III [RCB-III] in luminal tumors, ranked by the difference between posterior probabilities at optimal exam cutoffs. B.** Performance of individualized imaging parameters, performed either before or during neoadjuvant chemotherapy (NACT), for the diagnosis of residual cancer burden-III [RCB-III] in nonluminal tumors, ranked by the difference between posterior probabilities at optimal exam cutoffs.

| Imaging Modality | Test | Moment performed | Cutoff | Posterior probability of RCB-III if above cutoff | Posterior probability of RCB-III if below cutoff | Difference of posterior probabilities |
|---|---|---|---|---|---|---|
| SWE | Shear wave maximum speed inside [cm/s] | b) After 2AC cycles | 9.06 | 33.06% | 17.81% | 15.25% |
| Ultrasound | largest diameter [mm] | a) Baseline | 27.8 | 33.85% | 21.31% | 12.53% |
| Ultrasound | Peak Systolic [cm/s] | a) Baseline | 12 | 34.78% | 23.12% | 11.65% |
| MRI | Largest diameter [mm] | a) Baseline | 30 | 32.86% | 21.21% | 11.65% |
| MRI | Largest height [mm] | b) After 2AC cycles | 12 | 30.43% | 19.94% | 10.49% |
| Ultrasound | End diastolic velocity [cm/s] | a) Baseline | 1.5 | 31.17% | 21.83% | 9.34% |
| Ultrasound | Resistive Index | b) After 2AC cycles | 0.82 | 33.90% | 25.48% | 8.42% |
| SWE | Shear wave maximum speed out [cm/s] | b) After 2AC cycles | 4.47 | 32.44% | 24.51% | 7.93% |
| Ultrasound | Peak Systolic [cm/s] | b) After 2AC cycles | 6.9 | 29.77% | 23.14% | 6.63% |
| Mammography | Largest diameter [mm] | a) Baseline | 36 | 32.15% | 26.01% | 6.14% |
| Ultrasound | End diastolic velocity [cm/s] | b) After 2AC cycles | 2 | 30.20% | 24.50% | 5.70% |
| Ultrasound | Systolic/diastolic ratio index | a) Baseline | 5.79 | 30.85% | 25.26% | 5.59% |
| Ultrasound | Systolic/diastolic ratio index | b) After 2AC cycles | 5.71 | 31.58% | 26.20% | 5.38% |
| Mammography | Largest height [mm] | b) After 2AC cycles | 16 | 30.21% | 25.24% | 4.97% |
| Ultrasound | Pulsatility Index | a) Baseline | 1.64 | 30.24% | 25.40% | 4.84% |
| Ultrasound | Largest diameter [mm] | b) After 2AC cycles | 20.7 | 29.82% | 26.09% | 3.74% |
| Ultrasound | largest height [mm] | b) After 2AC cycles | 11.3 | 29.41% | 25.86% | 3.55% |
| Ultrasound | Pulsatility Index | b) After 2AC cycles | 1.55 | 29.94% | 26.42% | 3.53% |
| MRI | Largest height [mm] | a) Baseline | 28 | 29.87% | 26.66% | 3.21% |
| Mammography | Largest diameter [mm] | b) After 2AC cycles | 21 | 29.16% | 26.22% | 2.94% |
| MRI | Largest diameter [mm] | b) After 2AC cycles | 27 | 29.34% | 26.60% | 2.74% |
| Ultrasound | Largest height [mm] | a) Baseline | 17.3 | 28.57% | 26.98% | 1.59% |
| Ultrasound | Resistive Index | a) Baseline | 0.82 | 28.11% | 27.45% | 0.66% |

(*Continued*)

**Table 3.** (Continued)

| Imaging Modality | Test | Moment performed | Cutoff | Posterior probability of RCB-III if above cutoff | Posterior probability of RCB-III if below cutoff | Difference of posterior probabilities |
|---|---|---|---|---|---|---|
| Ultrasound | Shear wave maximum speed inside [cm/s] | a) Baseline | 9.83 | 27.58% | 28.11% | -0.53% |
| Imaging Modality | Test | Moment performed | Cutoff | Posterior probability of RCB-III if above cutoff | Posterior probability of RCB-III if below cutoff | Difference of posterior probabilities |
| MRI | Largest height [mm] | a) Baseline | 42 | 49.69% | 3.95% | 45.74% |
| MRI | Largest diameter [mm] | a) Baseline | 51 | 33.06% | 4.49% | 28.57% |
| SWE | Shear wave maximum speed out [cm/s] | b) After 2AC cycles | 3.76 | 30.51% | 3.84% | 26.67% |
| Ultrasound | Largest height [mm] | b) After 2AC cycles | 14.7 | 26.52% | 0.00% | 26.52% |
| Ultrasound | Largest height [mm] | a) Baseline | 26 | 27.29% | 3.49% | 23.80% |
| Mammography | Largest height [mm] | b] After 2AC cycles | 16 | 21.66% | 0.00% | 21.66% |
| Mammography | Largest diameter [mm] | b) After 2AC cycles | 40 | 26.22% | 8.44% | 17.78% |
| Ultrasound | Largest diameter [mm] | a) Baseline | 36 | 21.32% | 4.25% | 17.08% |
| MRI | Largest height [mm] | b] After 2AC cycles | 26 | 23.51% | 8.76% | 14.75% |
| Ultrasound | Largest diameter [mm] | b) After 2AC cycles | 30.6 | 20.50% | 7.28% | 13.22% |
| SWE | Shear wave maximum speed inside [cm/s] | b) After 2AC cycles | 7.85 | 16.04% | 7.03% | 9.01% |
| Mammography | Largest diameter [mm] | a) Baseline | 40 | 17.80% | 8.96% | 8.84% |
| Ultrasound | Shear wave maximum speed inside [cm/s] | a) Baseline | 9.4 | 15.46% | 6.82% | 8.65% |
| Ultrasound | Pulsatility Index | b) After 2AC cycles | 1.64 | 17.00% | 10.95% | 6.06% |
| Ultrasound | End diastolic velocity [cm/s] | b) After 2AC cycles | 2.2 | 15.90% | 9.85% | 6.05% |
| Ultrasound | Peak Systolic [cm/s] | b) After 2AC cycles | 8.7 | 9.85% | 15.90% | -6.05% |
| Ultrasound | Systolic/diastolic ratio index | b) After 2AC cycles | 4.57 | 16.50% | 11.09% | 5.40% |
| Ultrasound | Peak Systolic [cm/s] | a) Baseline | 10.3 | 14.27% | 10.55% | 3.72% |
| Ultrasound | End diastolic velocity [cm/s] | a) Baseline | 1.9 | 14.03% | 11.71% | 2.32% |
| Ultrasound | Pulsatility Index | a) Baseline | 1.74 | 11.39% | 13.58% | -2.19% |
| Ultrasound | Resistive Index | b) After 2AC cycles | 0.78 | 14.08% | 12.02% | 2.06% |
| Ultrasound | Resistive Index | a) Baseline | 0.79 | 12.39% | 13.58% | -1.19% |
| Ultrasound | Systolic/diastolic ratio index | a) Baseline | 4.67 | 12.39% | 13.58% | -1.19% |
| MRI | Largest diameter [mm] | b) After 2AC cycles | 31 | 13.32% | 12.26% | 1.06% |

MRI = magnetic resonance imaging; SWE = shear wave elastography; AC = adriamycin [doxorubicin] plus cyclophosphamide.

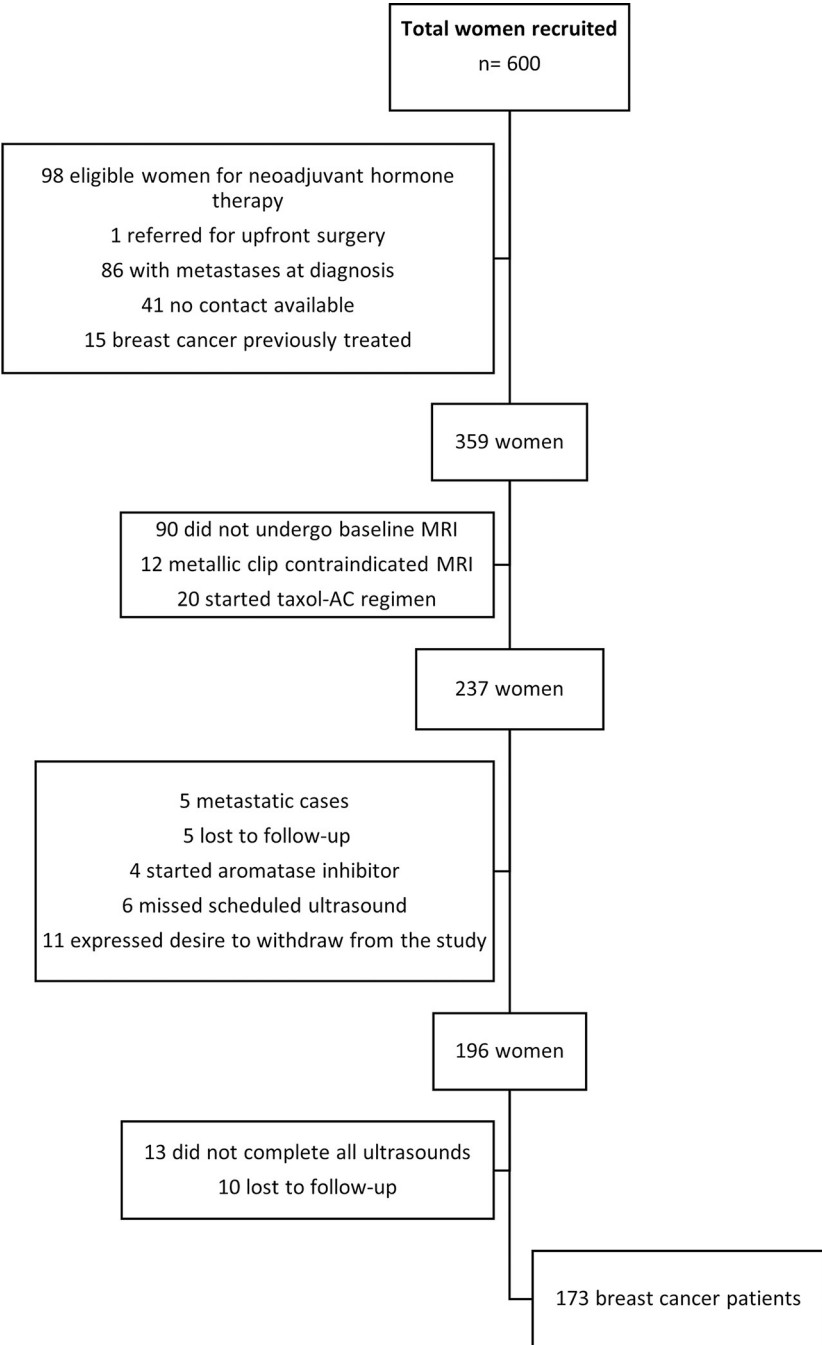

**Fig 1. Flowchart depicting the study design, patient attendance to follow-up consultations.**

above the threshold. Fagan's representations demonstrate that the prediction of pCR and RCB-III was more accurate for nonluminal tumors than for luminal tumors. Most exams performed better when conducted immediately before the start of NACT, with two exceptions that showed improved outcome prediction when performed mid-treatment, specifically after two cycles of NACT: 1) *Doppler* measurement of the end diastolic velocity for predicting pCR in nonluminal tumors and 2) SWE maximum speed inside the tumor for predicting RCB-III in luminal tumors.

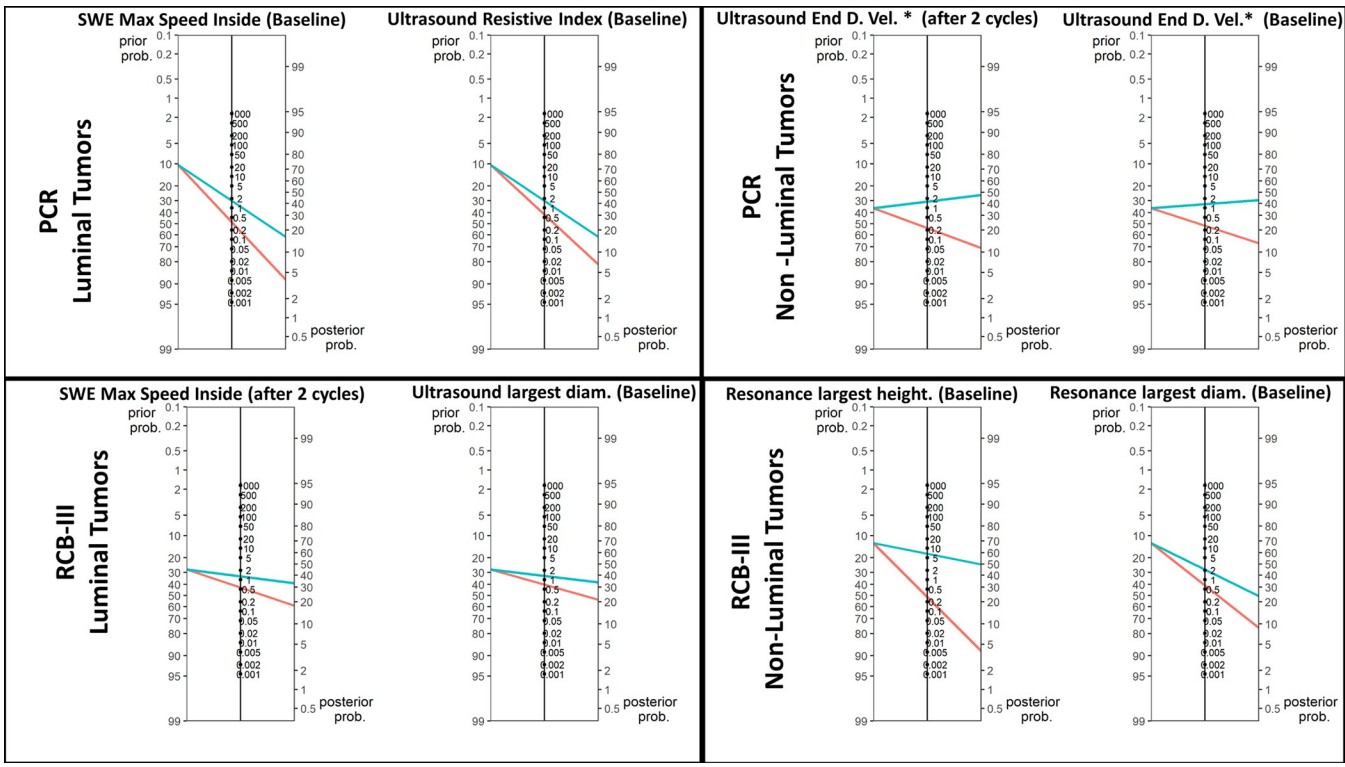

**Fig 2. Fagan's representation of prior and posterior probabilities in the best-case scenarios for predicting either pCR or RCB-III in luminal and nonluminal tumors.**

## Discussion

This prospective study aimed to assess the impact of pretreatment and mid-treatment exam parameters on the likelihood of a patient's response to NACT. The results of our study provide valuable insights into the use of simple interventions, such as mid-treatment Doppler ultrasound, in predicting treatment outcomes and adjusting management protocols for specific subsets of women.

We observed that the likelihood differences for NACT success or failure, as related to prior exam parameters, were more pronounced in women with nonluminal tumors than in those with luminal tumors. This can be attributed to the generally higher response rates of nonluminal tumors to NACT. The information obtained from mid-treatment tests can guide course corrections during NACT, particularly for women with triple negative or HER2 tumors who have access to new treatment options. Our study highlights the imaging modalities that can be pursued in such circumstances.

Specifically, our results demonstrate that women with nonluminal tumors and an end diastolic velocity above 1.9 cm (about 0.75 in/s) have a 35% higher chance of achieving a complete response to NACT than those with velocities below this threshold. Importantly, end diastolic velocities are easily measurable even by untrained sonographers using capable ultrasound equipment. While mid-NACT measurements yielded optimal results, measurements performed before the start of NACT still provided a good differential probability (approximately 29%) of a complete response.

In terms of predicting RCB-III, our study revealed that, similar to pCR, exams were more accurate for nonluminal tumors. However, for RCB-III, tumor size measurements showed

better discriminative power than functional measurements such as blood flow velocities or SWE measurements. This suggests that RCB-III is strongly associated with initial tumor burden, while pCR may be more influenced by tumor vascularization and the delivery of cytotoxic agents to the tumor's inner parts. Notably, MRI and MMG studies were superior to US in detecting the correlation between pre- and mid-treatment tumor sizes and RCB-III, indicating the higher precision of MRI in determining tumor burden.

The strengths of our study include a large sample size (n = 173) of NACT patients with fully annotated clinical data and comprehensive information on pre, mid-treatment, and post-NACT US, MRI and MMG. By comparing the most relevant exam parameters for predicting NACT response and identifying the optimal timing for measuring these parameters, we provide valuable insights. Our findings suggest that US morpho-functional parameters, such as Doppler velocimetry of tumors measured after two NACT cycles, can significantly contribute to discerning nonluminal tumors that are likely to respond to chemotherapy. It is important to emphasize that our US protocols can be easily reproduced, since the measurements of **ED, OS, MS, PI, and RI** are fully automated. The challenge lies in identifying the masses for which such parameters will be measured; however, the requisite ability can be acquired after a few months of training in a specialized imaging center and is far from requiring expert-level proficiency. Thus, we are confident that the measurements performed for our study can be easily reproduced in other centers, provided a reasonable amount of professional training is obtained, and capable equipment is available.

The relationship between blood flow parameters measured by Doppler velocimetry and response to NACT can be explained by increased exposure of tumor cells to chemotherapeutic agents in tumors with high blood flow and vice versa. Previous studies have suggested such a relationship [15], but prospective data on this relationship have been lacking until now. Interestingly, our study demonstrated that the relationship between Doppler velocimetry parameters and response to NACT was more pronounced in nonluminal tumors, while SWE measurements yielded better predictions of NACT response for luminal tumors.

It has been shown that lesion vascularity and stiffness decrease after NACT [16]. Doppler US was used to assess the number of tumor vessels after the first cycle of chemotherapy, which decreased in most cases. Tumors that responded well to NACT typically exhibited reduced vascularization. Tumors are known to primarily consist of small (less than 100 μm) and immature blood vessels with low blood flow velocity [17], which may help explain the predictive power of Doppler US findings for NACT response.

Our study highlights the leading role that US may play in the management of breast tumors during NACT. For predicting pCR after NACT, MMG parameters had limited value in the general prediction of tumor response. Changes in functional parameters, such as blood flow, velocity, and sound speed inside the tumor using US, demonstrated higher predictive value than morphological parameters. However, pre- and mid-treatment measurements of tumor diameters, especially using MRI, proved useful in predicting the risk of RCB-III, particularly for nonluminal cases. Further exploration is warranted to determine whether NACT protocols can be adjusted based on mid-treatment parameter changes to improve outcomes.

## Conclusions

Our study underscores the leading role of US in the management of breast tumors during NACT. Functional parameters, such as blood flow velocities and SWE measurements, demonstrated superior predictive value for pCR, while morphological parameters had limited value. However, pre- and mid-treatment measurements of tumor diameters, especially using MRI, proved useful in predicting the risk of RCB-III, particularly for nonluminal cases.

In conclusion, our study provides robust evidence supporting the use of mid-treatment Doppler US and other imaging modalities in predicting the response to NACT in breast cancer patients. These findings have implications for personalized treatment strategies and may contribute to improved outcomes in the management of breast cancer. Further exploration and validation of these findings are warranted to optimize NACT protocols and enhance patient care.

## Supporting information

**S1 File. Spreadsheet files containing data used in present analyses.**
(XLSX)

## Author Contributions

**Formal analysis:** Sophie Derchain.

**Investigation:** Rodrigo Menezes Jales, Maira Teixeira Dória, Carla Andries Cres Lyrio.

**Methodology:** Rodrigo Menezes Jales, Maira Teixeira Dória, Isabelle Melloni, Carla Andries Cres Lyrio, Carlos Menossi, Sophie Derchain.

**Project administration:** Livia Conz, Maira Teixeira Dória, Isabelle Melloni, Carlos Menossi.

**Resources:** Rodrigo Menezes Jales, Carlos Menossi.

**Supervision:** Livia Conz, Rodrigo Menezes Jales, Sophie Derchain, Luís Otávio Sarian.

**Validation:** Isabelle Melloni, Luís Otávio Sarian.

**Writing – original draft:** Luís Otávio Sarian.

**Writing – review & editing:** Livia Conz, Luís Otávio Sarian.

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
