## [Decision Letter · Decision Letter 0]

15 Feb 2024

PONE-D-23-36808Predictive Value of Ultrasound Doppler Parameters in Neoadjuvant Chemotherapy Response of Breast Cancer: prospective comparison with Magnetic Resonance and MammographyPLOS ONE

Dear Dr. Sarian,

Thank you for submitting your manuscript to PLOS ONE. After careful consideration, we feel that it has merit but does not fully meet PLOS ONE’s publication criteria as it currently stands. Therefore, we invite you to submit a revised version of the manuscript that addresses the points raised during the review process.

We look forward to receiving your revised manuscript.

Kind regards,

Daniele Ugo Tari, M.D.

Academic Editor

PLOS ONE

Journal Requirements:

Additional Editor Comments:

Dear Authors,

Your paper is well written and structured with only some minor revisions needed as requested by the reviewers.

Sincerely,

Reviewers' comments:

Reviewer's Responses to Questions

**Comments to the Author**

1. Is the manuscript technically sound, and do the data support the conclusions?

Reviewer #1: Yes

Reviewer #2: Yes

Reviewer #3: Yes

2. Has the statistical analysis been performed appropriately and rigorously? 

Reviewer #1: Yes

Reviewer #2: Yes

Reviewer #3: Yes

3. Have the authors made all data underlying the findings in their manuscript fully available?

Reviewer #1: Yes

Reviewer #2: Yes

Reviewer #3: Yes

4. Is the manuscript presented in an intelligible fashion and written in standard English?

Reviewer #1: Yes

Reviewer #2: Yes

Reviewer #3: Yes

5. Review Comments to the Author

Reviewer #1: The manuscript is interesting and well designed.

Introduction is good written.

Materials and methods are good designed

Results are good illustrated

Discussion is good written and include the conclusion of the study.

Reviewer #2: The study is well written, well structured and logical. I have a few questions about the part of methods:

1. Although the method of ultrasonic examination is explained in this paper, it is not detailed enough. As we all know, the repeatability of ultrasound examination is not good. If the final values of multiple ultrasound examination parameters mentioned in this paper are clear, such as ED, PS, MS, PI, RI, and lesion elasticity value, the results can be obtained. How to solve the problem of inspection repeatability in research?

2. In the pathological classification of RCB, only descriptive words are used to distinguish RCB-I, RCB-ii, and RCB-III. Is there any specific quantitative index to replace descriptive words as the classification standard?

Reviewer #3: The article is innovative. Simple and non-invasive interventions were used in the neoadjuvant process: ultrasound Doppler and shear wave imaging were used to evaluate the neoadjuvant treatment of intracavitary or non-intracavitary tumors, respectively, to observe the tumor prognosis.

1. The sum of the number of cases in line 13 of the method paragraph in the article is 172 cases, not 173 cases. Please give an explanation.

2. Are the Doppler indicators averaged in the value of non-intracavitary tumors?

3. Is the area of interest measured by shear wave completely covering the lump or taking small pieces within the lump?

6. PLOS authors have the option to publish the peer review history of their article (what does this mean?). If published, this will include your full peer review and any attached files.

Reviewer #1: No

Reviewer #2: No

Reviewer #3: No

---

## [Author Response · Author response to Decision Letter 0]

8 Mar 2024

Dr . Daniele Ugo Tari. M.D.

Academic Editor

PLOS ONE 

Subject: Submission of Manuscript – "Predictive Value of Ultrasound Doppler Parameters in Neoadjuvant Chemotherapy Response of Breast Cancer: prospective comparison with Magnetic Resonance and Mammography"

Conz, L; Jales, R.M.; Doria, M.T.; Melloni, I.; Lyrio, C.A.C.; Menossi, C.; Derchain, S.F.M.; Sarian, L.O.

Dear Dr. Tari,

I am pleased to submit the revised version of four manuscript entitled "Predictive Value of Ultrasound Doppler Parameters in Neoadjuvant Chemotherapy Response of Breast Cancer: prospective comparison with Magnetic Resonance and Mammography" for further consideration for publication in PLOS ONE.

 In this revised version, we have addressed each of the three reviewers’ concerns. In the following pages, you will find a detailed rebuttal of such concerns and a description of the modifications made to the original manuscript. 

We hope that the adjustments made enhance the manuscript’s suitability for publication in PLOS ONE. Regardless of the final decision, we are grateful to the time and effort invested in reviewing our paper. 

We look forward to your feedback.

Sincerely,

Luis Otavio Sarian

M.D., PhD, Full Professor - Department of Obstetrics and Gynecology, Division of Breast and Gynecologic Oncology

State University of Campinas (Unicamp), Campinas, São Paulo, Brazil

---

## [Decision Letter · Decision Letter 1]

8 Apr 2024

Predictive Value of Ultrasound Doppler Parameters in Neoadjuvant Chemotherapy Response of Breast Cancer: prospective comparison with Magnetic Resonance and Mammography

PONE-D-23-36808R1

Dear Dr. Sarian,

We’re pleased to inform you that your manuscript has been judged scientifically suitable for publication and will be formally accepted for publication once it meets all outstanding technical requirements.

Kind regards,

Daniele Ugo Tari, M.D.

Academic Editor

PLOS ONE

Additional Editor Comments (optional):

Reviewers' comments:

Reviewer's Responses to Questions

**Comments to the Author**

1. If the authors have adequately addressed your comments raised in a previous round of review and you feel that this manuscript is now acceptable for publication, you may indicate that here to bypass the “Comments to the Author” section, enter your conflict of interest statement in the “Confidential to Editor” section, and submit your "Accept" recommendation.

Reviewer #1: All comments have been addressed

Reviewer #3: All comments have been addressed

2. Is the manuscript technically sound, and do the data support the conclusions?

Reviewer #1: Yes

Reviewer #3: Yes

3. Has the statistical analysis been performed appropriately and rigorously? 

Reviewer #1: N/A

Reviewer #3: Yes

4. Have the authors made all data underlying the findings in their manuscript fully available?

Reviewer #1: Yes

Reviewer #3: Yes

5. Is the manuscript presented in an intelligible fashion and written in standard English?

Reviewer #1: Yes

Reviewer #3: Yes

6. Review Comments to the Author

Reviewer #1: The revised manuscript was good written and discussed. All reviewer comments have been addressed in the manuscript.

Reviewer #3: This study has implications for personalized treatment strategies and may contribute to improved outcomes in the management of breast cancer. In this study, ultrasound elastography and Doppler variables were used to evaluate neoadjuvant chemotherapy in breast cancer patients.

The US parameter with the highest power for predicting pathological complete response (pCR) was shear wave elastography (SWE) maximum speed inside the tumor at baseline. For nonluminal tumors, the end diastolic velocity

measured by US after two cycles of NACT showed the highest predictive value for pCR. Similarly, SWE maximum speed after two cycles of NACT had the highest discriminating power for predicting RCB-III in luminal tumors, while the same parameter measured at baseline was most predictive for nonluminal tumors.

7. PLOS authors have the option to publish the peer review history of their article (what does this mean?). If published, this will include your full peer review and any attached files.

Reviewer #1: No

Reviewer #3: No

---

## [Editor Report · Acceptance letter]

26 May 2024

PONE-D-23-36808R1 

PLOS ONE

Dear Dr. Sarian, 

I'm pleased to inform you that your manuscript has been deemed suitable for publication in PLOS ONE. Congratulations! Your manuscript is now being handed over to our production team.

Kind regards, 

on behalf of

Dr. Daniele Ugo Tari 

Academic Editor

PLOS ONE